Resource

# Dissecting morphogenetic apoptosis through a genetic screen in Drosophila

Audrey Barbaste[1],*, Sonia Schott[1],* , Corinne Benassayag[1,2] , Magali Suzanne[1,2]

**Apoptosis is an essential cellular process both in normal development and pathological contexts. Screens performed to date have focused on the cell autonomous aspect of the process, deciphering the apoptotic cascade leading to cell destruction through the activation of caspases. However, the nonautonomous aspect of the apoptotic pathway, including signals regulating the apoptotic pattern or those sent by the apoptotic cell to its surroundings, is still poorly understood. Here, we describe an unbiased RNAi-based genetic screen whose goal is to identify elements of the "morphogenetic apoptosis pathway" in an integrated model system, the *Drosophila* leg. We screened about 1,400 candidates, using adult joint morphology, morphogenetic fold formation, and apoptotic pattern as readouts for the identification of potential apoptosis-related genes. We identified 41 genes potentially involved in specific aspects of morphogenetic apoptosis: (1) regulation of the apoptotic process; (2) formation, extrusion, and elimination of apoptotic bodies; and (3) contribution to morphogenesis downstream of apoptosis.**

## Introduction

Apoptosis or programmed cell death is a fundamental cellular process. First described in 1972 as a clean way to eliminate cells (Kerr et al, 1972), apoptosis is recognized as an integral part of development since 1985, with the discovery of genes regulating this process during development in *C. elegans* (Ellis & Horvitz, 1986). Since then, apoptosis has been shown to be tightly regulated and controlled spatiotemporally during Drosophila development (White et al, 1994). Indeed, whereas apoptosis can be induced in response to stress signals to eliminate damaged cells, its role in morphogenesis must be precisely regulated. It is involved in numerous morphogenetic processes in both vertebrate and invertebrate models including digit formation, neural tube closure, and heart formation (Fuchs & Steller, 2011; Ambrosini et al, 2017). Morphogenesis appears to be a privileged

context to study apoptosis in an integrated system and in a natural environment because apoptosis is normally induced, without requiring any artificial trigger, and promotes tissue shape, offering an easy readout of perturbations of the apoptotic pathway.

Apoptosis is a stepwise process including cell shrinkage, blebbing, and fragmentation eventually leading to the extrusion and elimination of the remnant apoptotic bodies by phagocytosis (Kerr et al, 1972). At the cellular level, the apoptotic pathway converges on the activation of caspases, proteases that cleave a large number of proteins and thus orchestrate the progressive destruction of the cell. In cultured cells, these different steps rely on the activation of cell contractility, driven by the cleavage and activation of the kinase ROCK (Coleman et al, 2001). In an epithelial tissue, apoptotic cell dynamics start with an apical constriction independent of the actomyosin network, which relies on the destabilization of apical microtubules (Villars et al, 2022), followed by an apico-basal pulling force generated by an apico-basal cable-like structure of actomyosin (Monier et al, 2015), and the fragmentation of the cell into apoptotic bodies.

In addition, recent findings underline the importance of a nonautonomous role of apoptosis, regulating the density of apoptotic cells to ensure the maintenance of epithelial integrity (Valon et al, 2021) or stimulating compensatory proliferation (Fan & Bergmann, 2008). Apoptosis can also contribute to morphogenesis by modifying the surrounding tissue mechanically. It could lead, for example, to the creation of a pulling force accelerating epithelial sheet migration (Toyama et al, 2008), or participate in tissue folding through a local deformation that results in an increase in tissue tension and progressive deepening of the fold (Monier et al, 2015; Ambrosini et al, 2019; Roellig et al, 2022). Alternatively, it can influence tissue fluidity (Suzanne et al, 2010; Saw et al, 2017). However, if the mechanical impact of apoptotic cells on the neighboring tissue has been well described in a number of different contexts, the cellular and molecular mechanisms involved remain largely unknown.

A large number of screens have been performed, either in cultured cells (Seshagiri et al, 1999; Machuy et al, 2005; Chew et al, 2009; Lam et al, 2009; Yuan et al, 2016; Panganiban et al, 2019) or in epithelial tissues (Geuking et al, 2005; Kim et al, 2010; Fan et al, 2014;

[1]Laboratoire de Biologie Cellulaire et Moléculaire des Mécanismes du Contrôle de la Prolifération (LBCMCP), Centre de Biologie Intégrative (CBI), Université de Toulouse, CNRS, UPS, Toulouse, France [2]Molecular, Cellular and Developmental Biology unit (MCD), Centre de Biologie Intégrative (CBI), Université de Toulouse, CNRS, UPS, Toulouse, France

Correspondence: corinne.ben-assayag@univ-tlse3.fr; magali.suzanne@univ-tlse3.fr
*Audrey Barbaste and Sonia Schott are co-first authors

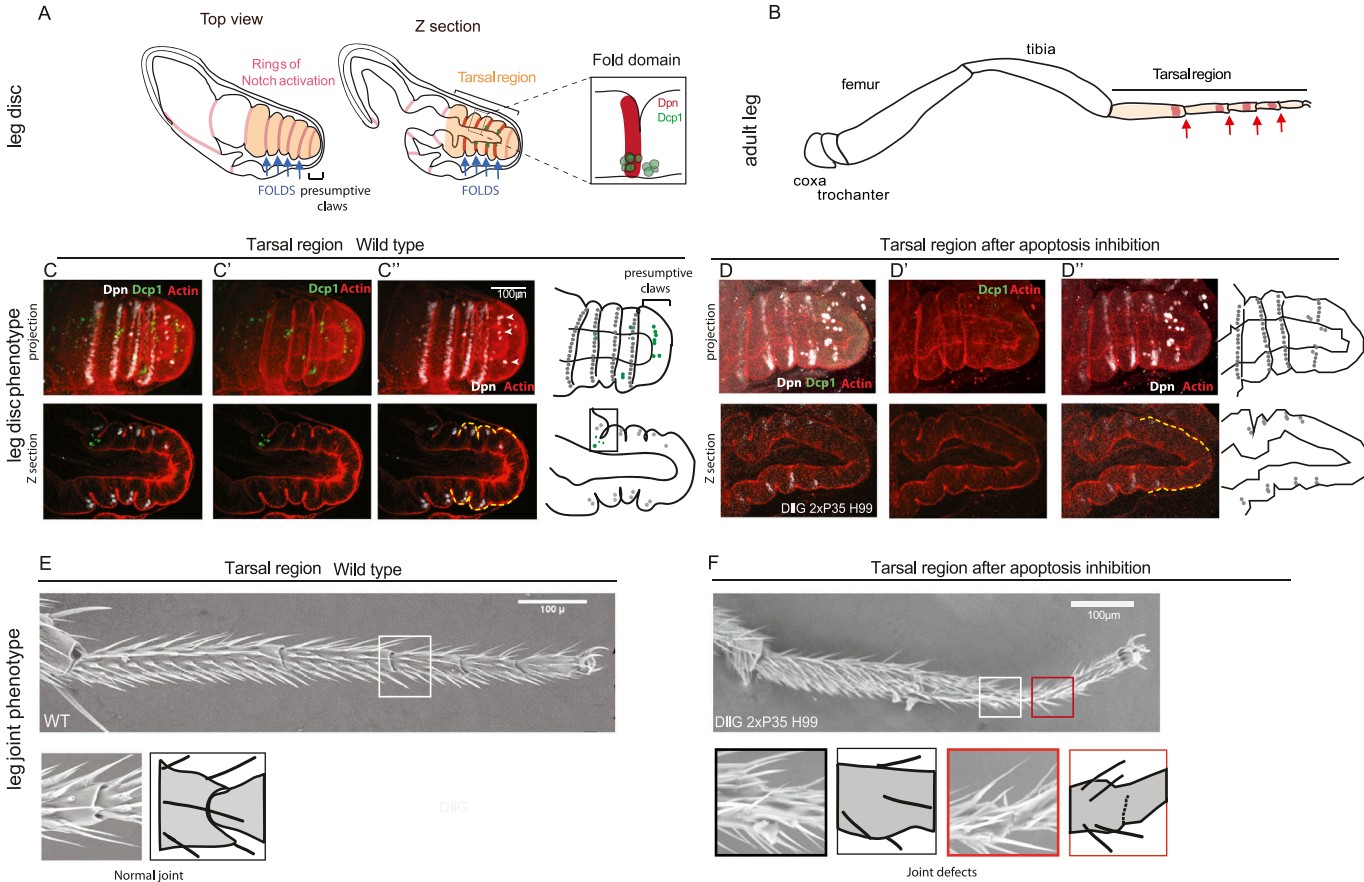

**Figure 1. A model to decipher morphogenetic apoptosis: the Drosophila leg.**
**(A)** Schemes of the developing leg disc showing a top view of the tissue with the ring of Notch activation in pink (left) and a z-section highlighting the tarsal region (more distal part) in which we note the presence of deadpan (Dpn) and apoptotic cells in the fold domains (blue arrows). The fold domain is shown at higher magnification on the right. **(B)** Scheme of the adult leg showing the different segments and highlighting the activation domains of Notch in red, proximal to each tarsal joint (red arrows). **(C, D)** Confocal images (3D reconstructions on top; z-sections below) showing the expression of Dpn (white), the presence of apoptotic cells labeled by anti-activated Dcp1 (green), and the F-actin in red to visualize the morphology of the tissue (the yellow dotted lines follow the disc contour, highlighting the presence or absence of folds) in a control (C) and in a Dll-Gal4 [EM212], UAS-p35; H99 background (D). Note that the apoptosis inhibition mainly affects the t4–t5 fold. Schematizations of the 3D (top) and z-section (bottom) are presented on the right. **(E, F)** Images from scanning electronic microscope of a control (E) and Dll-Gal4 [EM212], UAS-p35; H99 adult leg (F). Note that inhibiting apoptosis can affect joint formation to different degrees. Zoom of the joint region and corresponding drawings are shown below. Note the total absence of joints (left) or the partial fusion of the tarsus (right). In all figures, all the constructs (Gal4, UAS) or mutations (H99) are in heterozygosis. Scale bars are 100 $\mu m$.

Colin et al, 2015; Bejarano et al, 2021; Lebo et al, 2021). These screens, which generally involved the massive induction of apoptosis, allowed the identification of numerous genes involved in the apoptotic pathway, in particular, the different components involved in the cascade leading to caspase activation. Importantly, most of these components are expressed ubiquitously, permitting the fast elimination of any cell in response to a specific signal. This observation also highlights the need of a tight regulation of the apoptotic pathway to avoid spurious activation and underscores the importance of studying naturally induced apoptosis.

Importantly, none of the screens mentioned above addressed the question of the regulation, the cellular dynamics or the impact of naturally induced apoptosis in epithelial layers. Although interest in the dynamics of the epithelial apoptotic cell and its communication with the surrounding tissue is growing in the scientific community, an overview of the genes involved is still missing.

To better understand how programmed cell death contributes to epithelial dynamics at a multiscale level, we conducted an unbiased, RNAi-based silencing screen in a context of apoptosis-dependent morphogenesis, using Drosophila leg joint formation as a model. Our aim is to decipher the gene network involved in apoptotic-dependent morphogenesis from the regulation of the apoptotic pattern, to the orchestration of epithelial cell elimination, and finally the response induced in the surrounding cells.

## A model to decipher morphogenetic apoptosis: the Drosophila leg

To identify new genes involved in the "morphogenetic apoptosis pathway," we chose the formation of the Drosophila leg joint as a model. It has several advantages: it has a well-characterized apoptotic pattern, a well-described apoptotic cell dynamic, and an

easy read-out of apoptosis defect in the adult (Manjón et al, 2007; Monier et al, 2015).

In the developing leg imaginal disc, folds appear sequentially along the proximo–distal axis, dividing the tissue into different segments (Fig 1A). Fold formation depends on the activation of Notch (N) signaling in a row of cells in the distal part of each segment, forming a well-defined segmental pattern composed of eight concentric rings of Notch activation (see Fig 1A). The segmental activation of Notch defines tissue coordinates along the proximo–distal axis of the developing leg which are essential for the correct formation and positioning of the folds (de Celis et al, 1998; Rauskolb, 2001). Later during development, the different segments form the coxa, the trochanter, the femur, the tibia, and the five tarsal segments of the adult leg (Fig 1B and E). Each of these segments is bordered by flexible joints (see Fig 1B, red arrows, and inset in Fig 1E) that derive from the fold regions (Fig 1A, blue arrows) (Tajiri et al, 2010).

In the tarsal region (indicated in pale orange in Fig 1A and B), the N activation pattern can be visualized by the expression of Deadpan (Dpn) in four parallel rows of cells (in red in Fig 1A, Z section, and fold domain; in white in Fig 1C) and in neurons (Fig 1C", white arrowheads). Notch activation controls a number of cellular processes necessary to drive tissue remodeling. Among them, apoptosis has been shown to be mainly restricted to the presumptive fold domain (schematized in green in Fig 1A and shown in Fig 1C') where it actively contributes to tissue folding by the generation of a mechanical force in the depth of the epithelium. Apoptosis occurs in a stereotyped pattern in the tarsal region, and in the distal-most leg domain that will form the claws (Fig 1C').

In Notch mutants, the fusion of the different leg segments leads to a total absence of folds and joints (de Celis et al, 1998). Similarly, the inhibition of apoptosis in the distal part of the leg leads to fold and joint defects there (Fig 1D and F, respectively).

To inhibit apoptosis in the distal part of the leg via Gal4-driven expression of UAS-p35 lines, we first tested the DllGal4-MD23 driver used in previous studies and expressed in a domain that spread from mid-tibia to the claws (Manjón et al, 2007; Monier et al, 2015). In this genetic context, leg folds are strongly reduced, but this is accompanied by a strong reduction of Notch signaling (Fig S1A''') suggesting that this driver perturbs leg patterning. In addition, DllGal4-MD23 on its own shows a total deletion of the distal part of the leg disc at 30°C, revealing a strong defect in patterning (not shown). We thus decided to use the DllGal4-EM212 driver to direct the expression of p35. We observed variable fold defects in terms of penetrance and strength (as exemplified in Fig 1D and F), without affecting Dpn expression (Figs 1D and S1A' and A''). Indeed, distal fold and distal joint can be either missing, partially absent or normal (Fig 1D and F), even though leg patterning is well established. This variability, potentially because of the residual presence of some apoptotic cells, may show the robustness of this cellular process in the developing leg. Consistently, even when p35 is expressed in combination with the H99 deficiency that removes the three proapoptotic genes: *reaper*, *hid*, and *grim* (Fig S1A' and A''), some apoptotic cells can remain. Alternatively, it could indicate that other mechanisms are at play, compensating for the reduction of apoptosis.

Interestingly, we observed that inhibiting apoptosis also affects the formation of the arista, leading to the fusion of the distal part of the antenna (Fig S1B). This makes it feasible to screen simultaneously for adult phenotypes reflecting distal joint malformation both in the leg and the antenna.

### Identification of candidate genes involved in apoptosis-dependent morphogenesis through the characterization of leg joint defects

We then performed an unbiased screen crossing the DllGal4-EM212 driver with a collection of UAS-RNAi lines from the Bloomington Stock Center. We chose to use the Valium 10 collection of RNAis with known molecular function (excluding CG) because it is a large collection representing 10% of the genome. It was also the most efficient one and with fewer off targets at the time we started the screen. Finally, although this collection focused on genes expressed in neurons and has been described as potentially enriched in transcription factors, ions channels, and transporters (Ni et al, 2009), it seems to have no bias for any specific family of genes when compared with the *drosophila melanogaster* genes (Fig S2).

In a primary screen, the UAS-RNAi lines were crossed with the DllGal4-EM212 driver, the progenies were grown at two different temperatures (25°C and 30°C) and the adult phenotypes observed and classified. Initially, we used in parallel the RnGal4 driver, expressed in a large portion of the Dll domain (from tarsal segments T1–T4), to test for any driver-specific effect. We did this for the first rounds of RNAis tested (119 first RNAis from Table S3 tested in alphabetical order). Then, because the phenotypes were similar, we decided to use exclusively the DllGal4-EM212. Among the 1,418 RNAi tested, 45 induced lethality before enclosing as adults, precluding any further analysis. Another 181 lines, associated with morphological defects in the adult leg (at 25°C, 30°C or both), were classified in the following categories: (1) distal loss (corresponding to a total deletion of the distal part of the leg); (2) misshapen tarsus (corresponding to legs that are correctly differentiated along their proximo–distal axis, but show strong deformations); (3) tarsal fusions (which correspond to an absence of joints); (4) tarsal fusions combined with other defects or (5) strong morphological defect (important malformations not listed in the other categories) (Fig 2A and listed in Table S1). When possible, the phenotype has been confirmed with a second RNAi line (Valium 20; listed in Table S1). An example of each category is presented in Fig 2B, showing the phenotype associated with the inactivation of four representative candidate genes: *Inx2*, *mam*, *chico*, and *kibra*; an exhaustive list of the genes identified in the different categories is presented in Fig 2C. A global list of the lines tested, the genes affected, and the phenotypes observed is provided (Table S1), including defects observed in both the leg and the antenna. We noted a strong correlation between the phenotypes observed in the leg and in the antenna, with about 40% of the candidates showing tarsal leg fusion likewise presenting visible antennal fusion (Fig S1C). Finally, strong morphological defects were often associated with a fusion in the arista, suggesting that they could be apoptosis-related (Fig S1D).

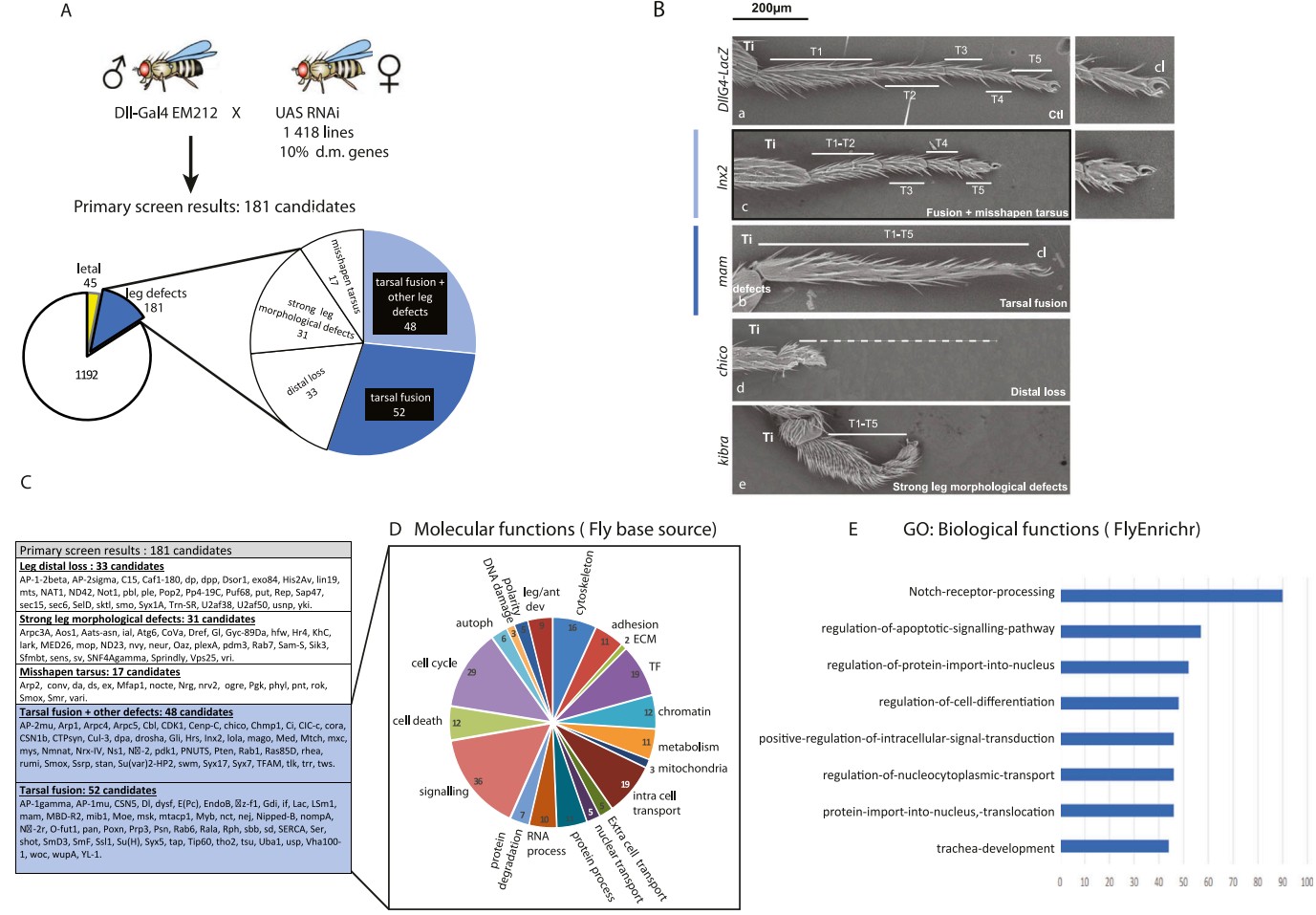

**Figure 2. Identification of candidates affecting adult leg joint formation.**
**(A)** Scheme showing the principle of the screen. **(B)** Representative examples showing images from scanning electron micrographs of the different types of adult phenotypes observed. Scale bar is 200 μm. **(C)** List of candidates identified through the primary screen. **(D)** Groupings of the candidates by different molecular functions. **(E)** Enrichment of the candidates for different biological processes using Flyenrich. Note the enrichment in genes related to the Notch and the apoptotic pathways.

To validate our approach, and because the screen was carried out blind (e.g., using numbered lines without knowledge of the target gene), we asked whether genes expected to give a phenotype, based on previous studies, have been selected in this screen. Interestingly, 40 genes identified as positive candidates based on the adult phenotype that are known to be involved in leg development, related to the Notch pathway, and/or involved in the apoptotic pathway (Table S2). We noticed that some known key apoptotic players such as *Dcp1* (caspase), *skl*, *rpr*, and *hid* (proapoptotic genes) do not give any tarsal fusion defects, which is coherent with the fact that mutants of these genes are viable (see Table S3). We then chose to focus our analysis on the 100 candidates whose inactivation leads more specifically to joint defects in the adult (either alone or in combination with another type of morphological defect), which correspond to 7% of the genes tested (highlighted in blue in Fig 2C and Table S1). No particular enrichment was observed regarding molecular functions as defined in Flybase (Fig 2D); in contrast, the strong enrichment for components of both the Notch

and the apoptotic pathways among the candidates selected from this primary screen, as judged using the Flyenrich classification, validates our approach (Fig 2E) (Chen et al, 2013; Kuleshov et al, 2016, 2019). Of note, the enrichment in Notch-related genes could also be due, at least in part, to the fact that the driver used is a Dll mutant that could show some genetic interaction with the Notch pathway.

## Identification of genes from the morphogenetic apoptosis network through a secondary screen

For these 100 candidates with leg joint defects in the adult, we then performed a secondary screen, looking for potential morphogenetic defects during prepupal leg disc development. As expected, for many of the candidates, gene inactivation led to fold absence or fold reduction (see the example of *PTEN* RNAi in Fig 3A and the pie chart on the right). Interestingly, inactivation of some genes leads to unexpected phenotypes such as fold deviation sometimes associated with fold reduction (for the example of *arpc*4 RNAi in Fig 3A,

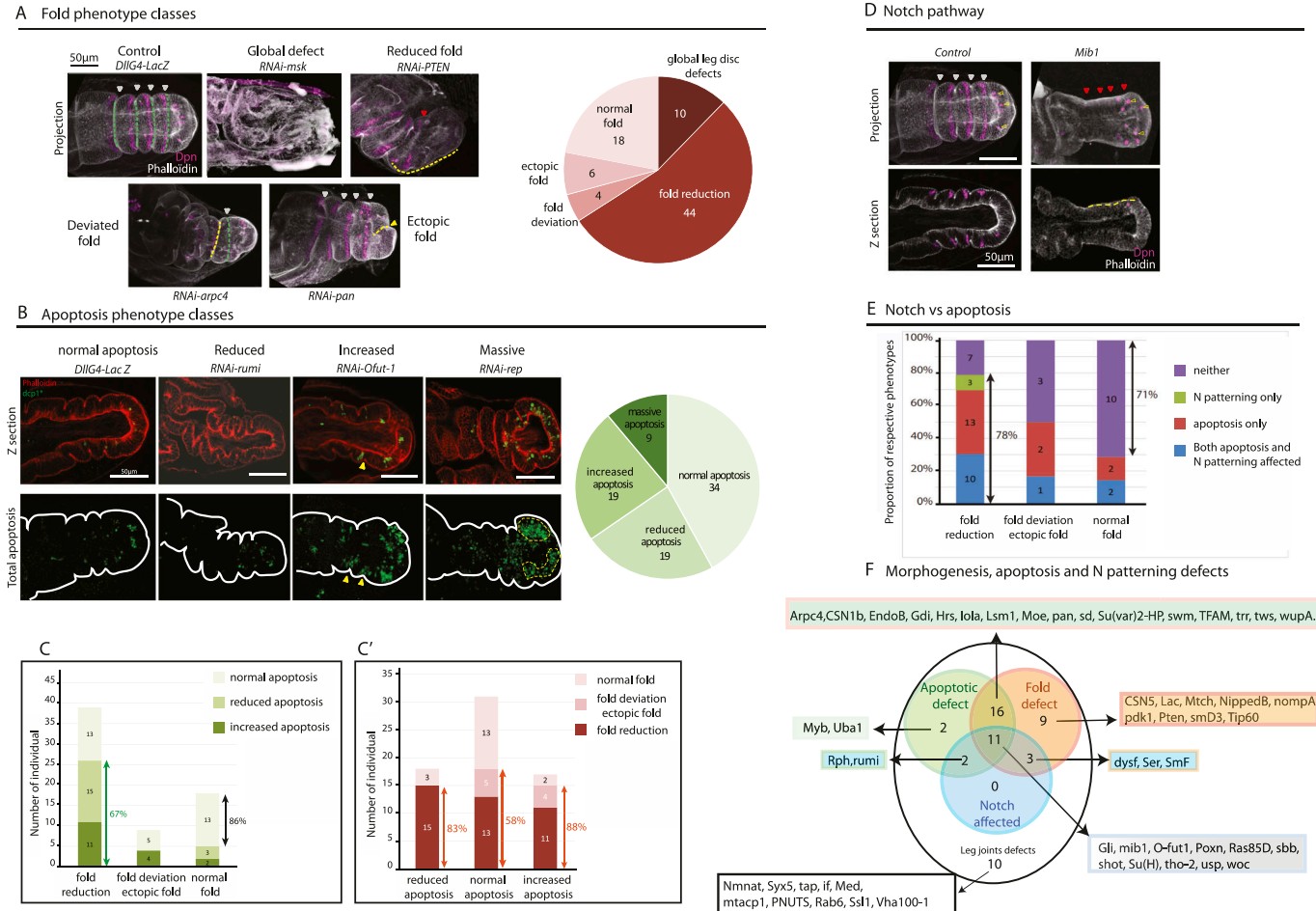

**Figure 3. Secondary screen based on fold formation, apoptotic pattern, and Notch activation defects in the developing leg disc.**
**(A)** Confocal images showing representative examples of morphological defects in the leg disc. F-actin is in white; Dpn expression in purple. Proportions of each class of defects are shown on the right (for 82 candidates showing clear results). The fourfolds are marked by white arrowheads and green dotted lines whereas fold defects are indicated by red arrowheads or yellow dotted lines. **(B)** Confocal images showing representative examples of apoptosis defects in the leg disc. Apoptotic cells are in green, F-actin in red. Proportions of each class of defects are shown on the right (for 82 candidates with clear results). Yellow arrowheads indicate the presence of apoptosis specifically in the fold domain, whereas nonspecific massive apoptosis is surrounded by a yellow dotted line. **(C, C')** Repartition of the different classes of apoptotic defects within the different classes of fold defects (C); repartition of the different classes of fold defects within the different classes of apoptotic defects (C'). **(D)** Confocal images showing representative examples of defects in Notch activation in the leg disc (*mib* RNAi). F-actin is in white, Dpn in purple. Of note, *mib1* inactivation specifically affected the pattern of Notch activation in the distal part of each segment, but not in neurons (yellow arrowheads not visible). The fourfolds are marked by white arrowheads, whereas missing folds are indicated by red arrowheads. The yellow dotted line shows the absence of fold. **(A, B, D)** Scale bars are 50 μm in (A, B, D). **(E)** Repartition of the number of candidates affecting Notch, apoptosis or both pathways within the different classes of fold defects. **(F)** Global repartition of the different classes of candidates. 53 candidates have been tested for each of the criteria and classified.

compare the orientation of the dotted lines in the *arpc4* RNAi sample versus the control) or to the formation of ectopic folds (see the example of *pan* RNAi in Fig 3A, yellow arrowheads; the respective frequencies are in the pie chart on the right). Finally, for a small number of candidates, we observed global defects of the leg disc, making it difficult to further describe the phenotype (see the *msk* RNAi in Fig 3A).

We next characterized the apoptotic pattern in these different mutant contexts. Interestingly, we could identify different categories of defects including reduced apoptosis (see the example of *rumi* RNAi, Fig 3B), normal apoptosis, increased apoptosis (more apoptotic cells are detected but still regionalized in the fold domains as indicated by yellow arrowheads, see the example of *Ofut-1*

RNAi) or massive apoptosis (more apoptotic cells are detected and the regionalization is lost, see the example of *rep* RNAi, Fig 3B, where zones of massive apoptosis are outlined). The repartition of the different classes of apoptotic defects reveals a large proportion of normal apoptosis, whereas reduced or increased apoptosis is observed in about half of the candidates (Fig 3B, pie chart on the right).

We next characterized the distribution of apoptotic defects in the different classes of fold defects. Although most of the candidates showing normal fold have a normal apoptotic pattern (86%, Fig 3C), apoptosis is frequently altered for the candidates showing a specific fold defect (67%, Fig 3C), as expected. In addition, the defects associated with reduced folds correspond mainly to (1) reduced apoptosis or (2) increased apoptosis, which may correspond

to an accumulated defect in the execution of the apoptotic program leading to the non-elimination of apoptotic debris. This enrichment of apoptotic defects when folds are reduced validates our methodology as a means to identify genes involved in apoptotic regulation or cell dynamics. Surprisingly, reduced apoptosis can also be detected in contexts where folds are normally formed (three candidates, graph Fig 3C), suggesting that apoptosis has to be strongly reduced or totally eliminated to observe a morphological defect. Finally, we noted that fold deviation and ectopic folds are associated with normal or increased apoptosis, suggesting that the deviated phenotype is not because of a reduction of apoptosis (Fig 3C).

Reciprocally, looking at the distribution of fold defects within the different categories of apoptotic classes, we observed that a large proportion of candidates show reduced folds when apoptosis is reduced (83%, Fig 3C'). We have also identified a number of candidates showing a normal pattern of apoptosis, but fold defects (58%, Fig 3C'). This category is particularly interesting because it could correspond to genes involved in fold and joint formation downstream of the apoptosis process, potentially including factors involved in force transmission in the leg tissue. Of note, increased apoptosis frequently leads to fold defects (88%, Fig 3C'). Although these defects include ectopic and deviated folds, they manifest most often as reduced folds, suggesting that defects in apoptotic cell elimination could impact morphogenesis.

To single out the genes involved in regulating apoptosis, we then decided to test the potential impact of reducing apoptosis on the phenotypes observed either in the adult or in the leg disc in the different mutant contexts. As expected, reducing the global level of apoptotic cells using the H99 deficiency accentuates leg joint and arista defects in adults (respectively, 32% and 51%, Fig S3A) and fold defects in the leg disc (38%, Fig S3B and Table S1).

We next tested whether the positional information was perturbed in these mutant contexts. Our main objective was to identify genes involved downstream of the establishment of positional information in the leg disc, however, our initial unbiased screen did not allow us to discriminate mutant contexts affecting early steps of leg development from genes involved specifically in fold morphogenesis. To determine whether developmental patterning was correctly established in mutant contexts, we used Notch activation as readout. As expected, given the role of Notch signaling in fold and joint formation, a number of candidates did show a defect in the segmental pattern of Notch activation as illustrated by the inactivation of *mind bomb* (*mib*), a well-known component of the Notch pathway (Fig 3D; note the absence of folds, indicated by red arrowheads, and the absence of Dpn).

We next analyzed, for each type of fold defect, the distribution of candidates affecting Notch signaling, apoptotic pattern or both. We observed that a large proportion of the candidates showing a reduction of folds was affected for the apoptotic and/or the Notch activation pattern (78%, Fig 3E), whereas normal folds strongly correlated with normal Notch activation and normal apoptosis (71%, Fig 3E). However, whereas around 35% of the candidates showing fold defects are affected for Notch activation, apoptosis was altered in about 70% (Fig S3C). This preferential effect on apoptosis indicates that our selected candidates were indeed enriched in genes involved in the morphogenetic apoptosis pathway as predicted.

Finally, to have a global overview of the candidates, we plotted the respective overlaps of the different categories of defects regarding Notch activation, apoptosis pattern, and fold formation (Fig 3F). In the graph, we can clearly see that all the candidates affecting Notch signaling showed either apoptotic defects or fold defects. We can also notice that most of the candidates affecting apoptosis (green circle) showed fold defects (orange circle). We note however that two candidates affecting both Notch and the apoptotic patterns did not show any fold defect. This observation suggests the potential existence of other mechanisms involved and/or that Notch and apoptosis need to be strongly reduced to show a phenotype. Finally, a small number of candidates only affect joint formation (10 leg joint defects), suggesting that they play a late, post-folding role in leg development. To better characterize candidates specifically involved in the morphogenetic apoptosis pathway, we decided to focus the rest of our analysis on the candidates showing fold defects (including various apoptotic and Notch activation patterns) that are not associated with strongly deformed tissue and/or massive apoptosis.

### Dissecting the morphogenetic apoptosis pathway

Interestingly, among the candidates perturbing fold formation (Fig 4), we identified a family of genes involved in different aspects of this apoptotic dependent process, including (a) reduced apoptosis, which suggests their role in apoptotic induction, (b) normal apoptosis, which suggests a potential role in downstream transmission in the apoptotic signals, and (c) increased apoptosis, which suggests either a misregulation of apoptosis or a defect in apoptosis orchestration (detailed below). These different apoptotic classes show no particular enrichment in any molecular function (not shown). Each of them has been selected initially based on the presence of joint defects in the adult leg upon their inactivation (phenotypes are shown in Fig S4).

***Induction: genes involved upstream of morphogenetic apoptosis***
The first class of candidates includes genes potentially involved in regulating the apoptotic pattern. In the leg disc, apoptosis is finely regulated as shown by its very stereotyped pattern. The leg disc is thus an excellent model to address apoptosis regulation. This is further supported by the fact that apoptosis is very difficult to inhibit totally in this tissue as shown by the residual presence of dying cells when only one copy of p35 is expressed (Fig S1A').

16 candidates (listed in Fig 6) showed a strong reduction in the number of apoptotic cells present in the tissue compared with the WT, although none of them totally inhibit apoptosis (maybe because of incomplete knock-down by RNAi). Representative examples are presented in Fig 4B (compare with the control in Fig 4A). They all show a strong variability in the morphological defects observed in the leg disc, similar to what was observed when apoptosis is inhibited. Some are expected to have an impact on apoptosis regulation: these include *Su(H)*, *Ser* and *Lola*, members of the Notch pathway (Yang et al, 2004; Reiff et al, 2019; Pérez et al, 2022), and *usp*, a gene shown to regulate apoptosis in response to ecdysone (Jiang et al, 2000; Daish et al, 2003; Cakouros et al, 2004) (see Fig 4B left panel). The identification of these candidates as potential upstream components of apoptosis induction in the

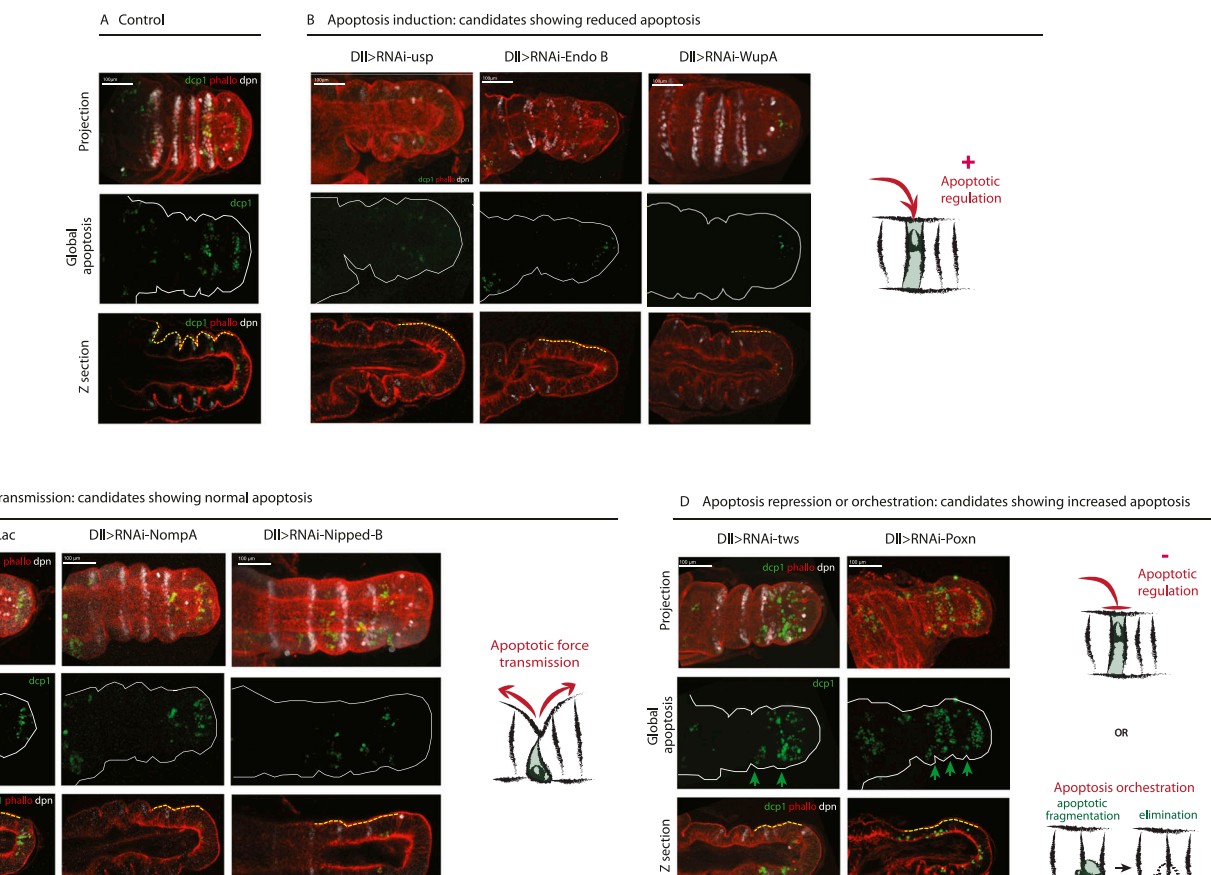

**Figure 4. Dissecting the morphogenetic apoptosis pathway.**
**(A, B, C, D)** Confocal images from the candidates showing fold defects showing the different apoptotic patterns (green) observed in representative examples of reduced apoptosis (B), normal apoptosis (C), and increased apoptosis (D), compared with the control (A). F-actin is in red. On the right of each class, a schematic representation depicts the putative gene function in the apoptotic process. Scale bars are 100 μm.

developing leg contributes to validate our approach. We further noticed that several candidates are involved in cell architecture, including membrane deformation (*EndoB*, see Fig 4B middle panel), cytoskeleton organization (*moesin*, *Gdi*, *Shot*, and *wupA*, see Fig 4B right panel), and adhesion complexes (*Nrx-IV*). Taken together, the identification of these structural components lends support to our working hypothesis that apoptosis could be regulated by mechanical signals in this developing tissue.

### Transmission: genes involved downstream of morphogenetic apoptosis

A number of candidates showed defects in fold formation despite normal patterns of morphogenetic apoptosis (listed in Fig 6). This suggested that the perturbation of fold formation is because of morphogenetic events taking place downstream of or parallel to the apoptotic process. Consistently, most such candidates do not affect Notch signaling (Fig 3F), further supporting a specific function downstream of tissue patterning. It was shown previously that apoptotic cells generate an apico-basal force in the developing leg, pulling on their neighboring cells. This results in increased local

tension and participates in the subsequent folding of the tissue (Monier et al, 2015; Ambrosini et al, 2019). However, how this force is regulated, and translated into an increase in tension in the neighboring cells, is still unknown. These candidates' identification could give us new insight into the molecular mechanisms involved in this mechanotransduction process. Among them, we identified candidates involved in cell trafficking (*AP1γ*, *Chmp1*) and in cell mechanics (*Pten*, *Arpc5*, *Lac*, and *nompA*, Fig 4C, left and middle panels), which open interesting new lines of investigation to decipher the mechanics of apoptotic force. We further identified genes potentially related to cell death and DNA repair-dependent processes (*Tip60*, *E(Pc)*, *CSN5*, *Mtch*, *Pdk1*, *Nipped*, Fig 4C right panel). Finally, a set of candidates is related to translation or transcription, which suggests their implication in the establishment of a downstream response driving tissue remodeling.

### Overabundance of apoptotic cell bodies

The last class of candidates showed an overabundance or accumulation of apoptotic bodies in the leg discs. Apoptotic corpses are present in abnormally high numbers possibly because more cells

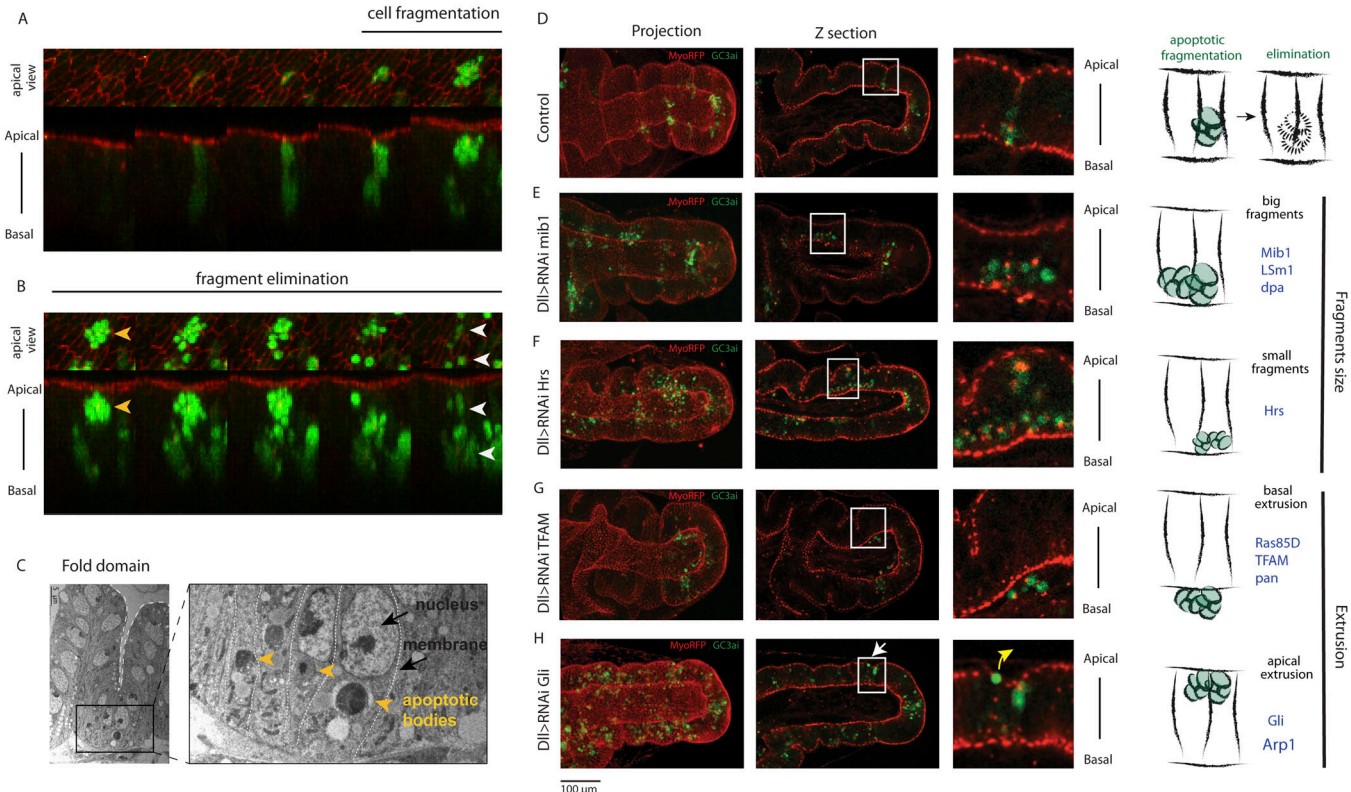

**Figure 5. Dissecting apoptotic cell fragmentation and extrusion.**
**(A, B)** Still image from a movie of a leg disc showing the dynamics of apoptotic cell fragmentation. Apical junctions are in red (followed by alpha-cat-RFPt), apoptotic cell is in green (GC3Ai). **(C)** Fold domain of a leg disc visualized by transmission electronic microscopy. Dotted lines indicate cell membranes. **(D, E, F, G, H)** Confocal images showing the different types of apoptotic corpse accumulation defects, including defects in fragment size (E, F), or fragment extrusion (G, H), compared with the control (D). Apoptotic bodies are visualized by the fluorescence of the apoptosis sensor GC3Ai (DllG4; UAS-GC3Ai; green); non-muscular myosin II (sqh-TagRFPt[3B]), in red, allows the visualization of the apical and basal surfaces of the epithelium. Scale bar is 100 $\mu$m.

are dying by apoptosis, because the corpses are not correctly eliminated, or both. We note however that apoptosis appears normally patterned, taking place in the presumptive fold, rather suggesting a defect in apoptotic bodies' elimination, although we cannot totally exclude that some of them correspond to an increase in the number of dying cells. Among them, we found genes involved in cell–cell junctions (*Gli*), the architecture of the actin cytoskeleton (*arpc4, Arp1*), and the intracellular degradation system (*CSN1B, Hrs, Uba1*), together with transcription and replication regulators (*TFAM, trr, Dpa, woc, LSm1, poxn*, and *tws*) and Notch-related genes (*Mib1, O-fut1*). Two representative examples are presented in Fig 4D affecting the N pathway (*Poxn*) or not (*tws*). The palette of different defects shown by this class of candidates opens new potential avenues of research (detailed below).

### Orchestrating apoptosis: dissecting apoptotic cell fragmentation

Among this last category with excessive apoptotic bodies, some candidate lines showed clear evidence of defects in the final steps of the apoptosis process, namely the fragmentation and/or elimination of the apoptotic fragments, leading to an accumulation of apoptotic bodies. We further analyzed their phenotypes using the apoptosensor GC3Ai, which allows visualizing the

apoptotic corpses and their fragmentation, and Dcp1antibody (Figs 5 and S5).

Epithelial apoptotic cell dynamics has been well described, in particular, using an early sensor of apoptosis that allows a 3D cell shape to be followed throughout the whole process (Schott et al, 2017). It starts with the reduction of the apical surface; the cell then contracts, pulling on the apical surface, forms blebs, and then fragments into apoptotic bodies. Because the cell first detaches on the basal side, the fragments form apically (Fig 5A). Then, they move basally and are phagocytosed most probably by the neighboring cells as suggested by the detection of apoptotic corpses in epithelial cells of the fold domain (Fig 5B and C).

Interestingly, we could identify different types of defects within the candidates showing an accumulation of apoptotic corpses. Indeed, some of them induced defects in the size of the apoptotic bodies. This is the case, for example, of *mib1*, an E3 ubiquitin ligase-regulating Notch signaling, whose inactivation leads to bigger fragments that appear to be stuck in the basal part of the epithelium (Figs 5E and S5B). Another example is *Hrs*, a component of the ESCRT complex involved in the degradation by the lysosome of ubiquitinated proteins, which, when inactivated, leads to a clear accumulation of small fragments in the most basal part of the epithelium (Figs 5F and S5C). These results suggest a role in the

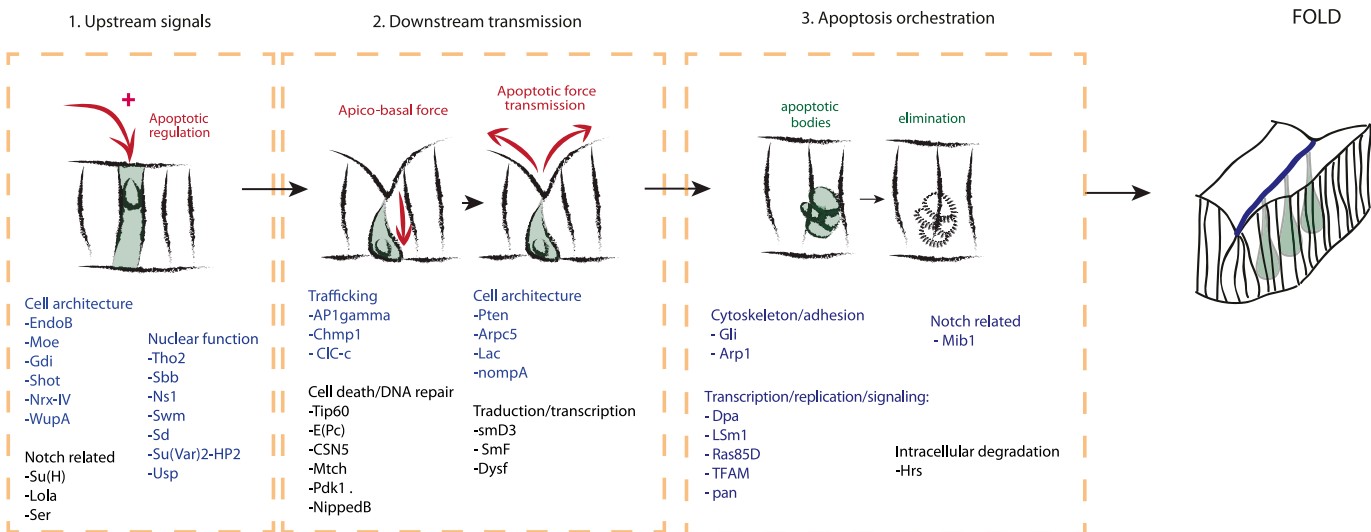

**Figure 6. Schemes summarizing the different types of candidates identified through this genetic screen, including: (1) new regulators of endogenous apoptosis, (2) genes involved in downstream transmission of the apoptotic signal, and (3) genes involved in apoptosis orchestration leading to fold formation.**

process of apoptotic cell fragmentation, where the degradation machinery has to be finely regulated in order for the apoptotic cell to be correctly fragmented and eliminated.

Some other candidates affect the extrusion of apoptotic fragments. This includes *TFAM*, involved in mitochondrial DNA transcription and replication, and *Gli*, a transmembrane protein component of the tricellular septate junction. TFAM knock-down leads to the accumulation of fragments below the basal part of the epithelium that is never observed in the control (Figs 5G and S5D), suggesting that the apoptotic bodies are expulsed from the epithelium but fail to be eliminated by phagocytosis. In contrast, apoptotic fragments are frequently observed in the apical part of the epithelium on inactivating *Gli*, suggesting that septate junctions could play a role in the elimination of the apoptotic bodies, preventing their apical expulsion from the epithelium and thus allowing their correct elimination on the basal part of the epithelium (Figs 5H and S5E). This phenotype is rare (only two candidates showed such defects) and has never been observed in a WT context, suggesting that apical extrusion is not efficient for the correct elimination of the apoptotic bodies. Of note, defects in the phagocytosis of apoptotic bodies could be either cell-autonomous (defects in the apoptotic cells themselves) or nonautonomous (defects in the phagocytic cells, which are most probably the neighboring epithelial cells).

Altogether, the panel of phenotypes described above highlights the complexity underlying the elimination of cell debris. These observations indicate that apoptotic cell fragmentation has to be finely executed to ensure the correct elimination of apoptotic corpses.

## Discussion

We have described here an unbiased genetic loss-of-function screen, the first seeking to identify genes playing a role in apoptosis-

dependent morphogenesis (Fig 6). The screen, which yielded a number of previously unknown actors in the process, was based on the initial identification of leg joint defects in the adult, followed by the detailed characterization of fold, cell death, and patterning defects in the developing leg. We were able to differentiate between genes affecting tissue patterning or specifically downstream processes related to apoptosis regulation, execution or force transmission.

Of note, a number of phenotypes came as a surprise. The first surprise was the observation of phenotypic variability in several mutants regarding the orientation of the folds. This has been observed in four mutant contexts including *arpc4*, *arpc5*, *Chmp1*, and *CSN5*. Based on these results and the identification of two components of the Arp2/3 complex, *arpc4* and *arpc5*, a recent study of the function of this complex in fold robustness has demonstrated the importance of a directional bias in transmitting apoptotic force to ensure stereotyped fold formation (Martin et al, 2021).

The second class of unexpected phenotypes concerns the different types of defects related to orchestrating apoptosis or eliminating apoptotic cell debris (Fig 5). Some of them regulate the size of the apoptotic bodies and are potentially involved in the regulation of apoptotic force, which contributes to cell fragmentation as shown in chicken embryos (Roellig et al, 2022). Others are potentially involved in fragment extrusion or phagocytosis. We noted that for most of them, the accumulation of apoptotic bodies is associated with defective folds, strongly suggesting that the correct elimination of cell debris can have an impact on morphogenesis.

Finally, some genes show a leg joint phenotype in the adult but no phenotype in the leg disc, neither for fold formation nor for the observed apoptosis pattern. Among them, five candidates are involved in intracellular degradation (*Vha100-1*, *Rab6*, *Cul-3*, *Nmnat*, *Cbl*), suggesting a possible degradative function in late pupal stages when these cells undergo extensive shape changes while secreting from the cuticle (Tajiri et al, 2010).

Overall, this study in an integrated model, going from single-cell dynamics to tissue morphogenesis, led to the identification of a number of interesting new potential avenues of study concerning apoptosis regulation, mechano-transduction of apoptotic force and apoptotic cell elimination. Among the candidates identified in this screen, some may have a specific function in regulating developmental apoptosis, whereas others will have a general function in a variety of contexts. Interestingly, on top of its important function during development, apoptosis is well known to play important roles in different pathological contexts including neurodegenerative disorders and cancer (Mattson, 2000; Igney & Krammer, 2002; Radi et al, 2014). Indeed, mutations in the apoptotic pathway are frequently found in tumors and some anti-tumoral compounds have pro-apoptotic properties leading to the common view of apoptosis as an antitumoral process (Debatin, 2004; Xu et al, 2011). However, surprisingly, apoptosis was found recently to exert a pro-tumoral role, by stimulating tumor growth (Huang et al, 2011; Ford et al, 2015; Pavlyukov et al, 2018), highlighting the importance of deciphering the growing perceived complexity of this cellular process. This screen should help open new directions of investigation regarding this new pro-tumoral function of apoptosis.

# Materials and Methods

### Drosophila stocks

Dll-Gal4 EM212 and Dll-Gal4 MD23 were gifts from G Morata's laboratory. UAS-GC3Ai has been described previously (Schott et al, 2017). The deficiency Df(3L)H99 (BDSC_1576), that removes the three proapoptotic genes hid, reaper, and grim, was a gift from H Steller (Rockefeller University, USA). The sqh-TagRFPt[3B] line is a knock-in line described in Ambrosini et al (2019). UAS-alphaCatenin-TagRFP was a gift from K Sugimura (iCeMS, Japan). UAS-p35 BL5072 and 5073 and 5074 (UAS-P35 on the chromosomes 2, 3, and X, respectively), RnGal4 BL7405: w; rnGAL4/ TM3Ftz^LacZ Ser Sb and UAS-RNAi lines (from the Valium 10 and 20 collections) were obtained from Bloomington Drosophila Stock Center (BDSC). We constructed for this analysis: w; Dll-Gal4 EM212; Df(3L)H99/ T(2–3) CyO; TM6b Hu, Tb and 2XP35: w; UASP35(2); UASP35(3).

### Immunofluorescence

To limit the intrinsic variability in the apoptotic pattern, imaginal leg discs were dissected 2 h after pupal formation in PBS 1X, then washed in PBS 1x, 0.3% triton x-100 and BSA 1% (BBT), then incubated overnight at 4°C with primary antibodies diluted in BBT. After the primary antibody, the samples were washed for 1 h in BBT, incubated for 2 h at room temperature with secondary antibodies diluted in BBT, washed with PBS 1x, 0.3% Triton X-100 for 1 h, and mounted in Vectashield (Vector Laboratories), using a 120-$\mu$m deep spacer (Secure-SealTM from Sigma-Aldrich) to preserve tissue morphology.

Anti-Dcp1 (Cell Signaling Technologies) and anti-Dpn (ab195172; Abcam) were both used at a dilution of 1/100. Phalloidine–Rhodamine (Thermo Fisher Scientific) used to stain F-actin was diluted at 1:200.

### Confocal imaging

Acquisitions were done using a LSM880 confocal microscope equipped with GaAsP and Airy detectors, and a Plan-Apochromat 40x/NA 1.3 Oil DIC UV-IR M27 objective. Post-acquisition treatment was done using Image J and Adobe Photoshop.

### Adult phenotypes

#### Candidate selection
Candidates were selected when at least 50% of the individuals were showing a phenotype. At least 20 individuals were observed for each genetic context and each temperature.

#### Surface electronic microscopy
Frozen adult structures were attached to a double-side tape and imaged with a TM1000 Hitachi scanning electron microscope.

#### Hoyer's
Adult legs were directly mounted in Hoyer's medium and imaged with a Nikon SMZ18 macroscope.

### Transmission electron microscopy

Prepupae (1–2 h APF) and pupae (3–4 h APF) were opened and fixed for 2 h at room temperature with 2.5% glutaraldehyde and 2% paraformaldehyde in 0.08 M cacodylate buffer containing 0.05% $CaCl_2$ (pH 7.2, EMS). Leg discs were then dissected and postfixed at room temperature with 1% $OsO_4$ in the same buffer. Samples were treated for 1 h with 1% aqueous uranyl acetate and were then dehydrated in a graded ethanol series followed by 100% acetone. Samples were flat embedded in EMBed-812 resin (EMS). After 48 h of polymerization at 60°C, ultrathin sections (80 nm) were mounted on singleslot Formvar-coated copper grids. Sections were stained with Uranyless (Delta Microscopies) and 3% Reynolds lead citrate (Chromalys). Grids were examined with a TEM (Jeol JEM-1400) at 80 kV. Images were acquired all along the PE in each section using a digital camera (Gatan Orius).

### Enrichment analysis using FlyEnrichr

The length of the bar represents the significance of the specific gene-set or term (combined score). This combined score is computed by taking the log of the $P$-value from the fisher exact test and multiplying that by the Z score of the deviation from the expected rank. All the presented biological functions are significant. For Notch receptor processing $P$-value = $7.91 \times 10^{-12}$ and for the regulation of apoptotic signaling pathway $P$-value = $7.62 \times 10^{-14}$ were calculated with the fisher exact test.

# Supplementary Information

# Acknowledgements

We thank Luisa di Stefano and Bruno Monier for their constructive comments on the article, David Cribbs for proofreading the article. M Suzanne's laboratory is supported by grants from the Agence Nationale de Recherche (ANR, PRC AAPG2021, CellPhy) and from the Association pour la recherche contre le cancer (ARC, Programme Labélisé AAP2020, ARCPGA12020010001154_1591). A Barbaste salary was supported by a grant from the European Research Council (ERC) under the European Union Horizon 2020 research and innovation program (grant number EPAF: 648001).

## Author Contributions

A Barbaste: formal analysis, investigation, visualization, and methodology.

S Schott: formal analysis, investigation, visualization, and methodology.

C Benassayag: conceptualization, data curation, formal analysis, supervision, validation, investigation, visualization, methodology, and writing—original draft, review, and editing.

M Suzanne: conceptualization, supervision, funding acquisition, project administration, and writing—original draft, review, and editing.

## Conflict of Interest Statement

The authors declare that they have no conflict of interest.

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
