## [Reviewer comments · Life Science Alliance]

Life Science Alliance

Dissecting morphogenetic apoptosis through a genetic screen in *Drosophila*

Audrey Barbaste, Sonia Schott, Corinne Benassayag, and Magali Suzanne

DOI: <https://doi.org/10.26508/lsa.202301967>

Corresponding author(s): Magali Suzanne, Centre de Biologie Integrative and Corinne Benassayag, Centre de Biologie Integrative

Review Timeline:

Submission Date:	2023-02-03
Editorial Decision:	2023-03-06
Revision Received:	2023-06-05
Editorial Decision:	2023-06-26
Revision Received:	2023-07-12
Accepted:	2023-07-13

Transaction Report:

March 6, 2023

Re: Life Science Alliance manuscript #LSA-2023-01967-T

Dr. Magali Suzanne
Centre de Biologie Integrative (CBI)
118 route de Narbonne
Toulouse 31062
France

Dear Dr. Suzanne,

Thank you for submitting your manuscript entitled "Dissecting morphogenetic apoptosis through a genetic screen in *Drosophila*" to Life Science Alliance. The manuscript was assessed by expert reviewers, whose comments are appended to this letter. We invite you to submit a revised manuscript addressing the Reviewer comments.

Thank you for this interesting contribution to Life Science Alliance. We are looking forward to receiving your revised manuscript.

Sincerely,

B. MANUSCRIPT ORGANIZATION AND FORMATTING:

Reviewer #1 (Comments to the Authors (Required)):

In this manuscript, the authors performed a RNAi screen to screen for genes that control morphogenetic apoptosis during formation of the leg joint of *Drosophila melanogaster*. They screened more than 1,400 genes (about 10% of the total genome) and identified 182 candidate genes. These were further characterized and grouped into distinct classes that affect specific aspects of morphogenetic apoptosis such as regulators of apoptosis, extrusion and elimination of apoptotic bodies, and acting downstream of apoptosis.

This is a very descriptive study, but the analysis of the candidates was done in a very rigorous manner. In the beginning of the manuscript, leg joint development is well explained which made it easy to follow the mutant analysis. The data are well presented and the conclusions are in line with the data. It will be very interesting in future work to uncover the specific molecular function of these candidate genes for morphogenetic apoptosis.

I have a few questions and suggestions.

1. Which criteria were applied to select the 1,418 genes to be screened for leg joint formation?
2. If I understood the screen and the follow-up analysis correctly, only one RNAi stock was used per gene. Did the authors confirm their findings with additional independent RNAi stocks of the candidate genes? The use of multiple RNAi lines per gene or even better conditional CRISPR/Cas9 knockouts (when available) is standard in *Drosophila* experimentation.
3. There is some inconsistency in the number of candidate genes: in the text there are 182 candidate genes, in Figure 2A, it's 181 and in Figure 2C, it's 226 candidates. Please clarify and correct where necessary.
4. While the manuscript is well written, I recommend a spell check and proof-reading by a native English speaker. There are several typos and grammatical errors in the text and the figures. For example, in Figure 3 the authors used the word "normale" which does not exist in the English language. In the text, they used the word "raw" instead of "row". There are more examples which I won't list, because the authors didn't provide page and line numbers.

Reviewer #2 (Comments to the Authors (Required)):

This work by Barbaste and Schott et al., reports the results of a genetic screen to uncover genes involved in *Drosophila* leg development. The screen was focused on finding genes related to joint development, a process that relies on patterned apoptosis. The screen is well designed, has a very useful readout and more importantly uncovers interesting genes involved not only in induction of apoptosis, but also in mediating the morphogenetic rearrangement downstream of the apoptosis process itself, among other related processes. This is a very useful in vivo model to study apoptosis and how it influences development. The hits identified will be of interest to a wide community as they add on the in vivo regulators of morphogenetic apoptosis. There are some points that would nevertheless strengthen the work and its conclusions:

1. The authors do not mention how penetrant are the phenotypes that they report. It would be important to mention how frequent was a phenotype for it to be considered as representative for a given knockdown condition, even if this was only estimated empirically.
2. The authors use a distal-less driver to knock down expression of target genes. According to Flybase, the EM212 driver is also an allele of *dll* and therefore, it not only drives gene silencing of the given RNAi lines, but it also provides a sensitized background. While this does not reduce the value of the screen or of the hits they uncover (in fact is a nice strategy to increase the number of hits) it could be interesting to confirm some of the results with an independent driver (if available). Related to this point, the authors report a great number of genes related to the Notch signaling pathway as hits for their screen and argue that this is due to the central relevance of N pathway in regulating joint development. However, *Dll* seems to regulate *Ser* expression in the legs (Grace Panganiban, John L. R. Rubenstein, 2002) so there could indeed be a genetic interaction with the driver enriching for Notch pathway components. This scenario could at least be mentioned in the discussion section.

3. Figure S3 shows some leg phenotypes in bright field images. Can the authors include a control leg in bright field to have a direct point of comparison?
4. Same figure, some legs do not have hairs (Nipped-B) or have excessive number of hairs (TFAM). This again points for Notch-related phenotypes. It would be useful if these categories could be added and discussed.
5. Many graphs do not have a title for the y-axis.
6. In section 2, when the authors describe how they classify the phenotypes, it would be good to give more detailed explanations of each category. Specifically what is "(2) strong morphological defect", is it any other deformation not included in numbers 3-5?
7. The authors show that in some conditions, apoptotic bodies are not properly engulfed and cleared. It would be interesting to check if the driver is expressed in macrophages/haemocytes or only in the epithelial tissue of the leg, which could hint to the cell type responsible of the phenotype (neighbouring epithelial cells vs hamocytes).
8. Regarding Arp2/3 in the orientation of the folds, it would be worth discussing how boundaries are maintained in place and whether the change of orientation in these knockdowns would be a consequence of misdefinition of the boundary or of a displacement of a correctly patterned boundary.
9. Same paragraph as above: The reference "Martin et al., 2021" is not in the reference list.
10. The fly drawings used in figure 2A seem to be cropped from another figure and have extra arrows or other characters. The authors may want to use the original vector images found in <https://sites.manchester.ac.uk/fly-facility/training/>
11. Please check the manuscript for typos or minor mistakes, some examples are:
 - Short summary: "We carried out" instead of "We realized"
 - Section 1, paragraph 2: ..."activation of Notch (N) signaling in a raw of cells"... row instead of raw.
 - Section 1, paragraph 3: "blue" instead of "bleu"
 - Figure 3B and elsewhere: "normal" instead of "normale"

I agree with reviewers 1 and 3 on the use of independent RNAi lines to validate some of the targets or discussing the possibility of off-targets. I also agree it would be important to mention how they define the list of 1,418 genes to be screened. The authors do mention it was an unbiased approach but do not explain how it was defined.

Reviewer #3 (Comments to the Authors (Required)):

1. This manuscript describes a screen for genes that affect *Drosophila* leg morphogenesis, with a focus on events regulated by Notch and apoptosis. They first show that expression of p35 in combination with heterozygosity for H99 leads to morphological defects with a lack of joint formation. They identify nearly 200 genes affecting this process, and categorize them by their effects on Notch and apoptosis. Interestingly, many of them have not been associated with apoptosis before, and the authors propose mechanisms by which cellular mechanics may affect activation of apoptosis or apoptotic cell elimination. All together this is a valuable resource paper that provides a useful starting point for future analysis. No major changes are needed but clarification on a number of points would improve the paper.

2. The conclusions are strongly supported by the data and no experiments are required. Improvements are suggested below.

Note: the tables cited in the text were not included with the documents for review so they could not be evaluated.

3. Minor comments:

- a. Are the apoptotic cells in the leg disc typically phagocytosed by macrophages or neighboring epithelial cells? More detail on this should be provided (or if it is not known that should be clearly mentioned) since the interpretation of the phenotypes will vary depending on this.
- b. On p. 4 in the text, it is mentioned that DllGal4 MD23 has a phenotype on its own at 30{degree sign} and refers to Fig S1A'. However this figure has p35 so it's not clear what the text is referring to.
- c. Are there extra arista branches in the antenna when apoptosis is inhibited? This phenotype has been previously described for hid loss of function or Diap1 overexpression.
- d. Please include precise genotypes. Is H99 heterozygous in all experiments? Please indicate if the Gal4 or p35 is heterozygous or homozygous.
- e. How were the candidate genes selected? It is important to understand if the frequency of "hits" is representative of the genome or if genes important for signaling or patterning were pre-selected.
- f. It is not surprising that Dcp-1 did not give a phenotype since Drice is the major effector caspase. Since mutants of Dcp-1, reaper and hid are viable, these could be examined for leg phenotypes to determine if they should have been identified in the screen. The meaning of the underlined portion of this sentence is not clear: "We noticed however that some known apoptotic key players such as Dcp1 (caspase), skl, rpr and hid (pro-apoptotic genes) do not give any tarsal fusion defects, showing here again the robustness of apoptosis pattern during joints morphogenesis."
- g. They use Dcp-1+ as a read out for apoptosis but Dcp-1 positivity/caspase activity is not always associated with apoptosis. It would be best to confirm it in some other way - for example with DAPI to show that pyknotic nuclei colocalize with the Dcp-1 label, or with TUNEL.
- h. It is not clear why they assume it is "accumulated apoptosis" (apoptotic cell elimination) rather than increased apoptosis. If

they can rule out increased apoptosis they should explain how this was determined.

i. The GFP sensor or Dcp-1 antibody may not be detectable in late phagocytosed corpses that have acidified. The mutants that don't show a defect in apoptosis could affect these late stages of phagocytosis that are not detectable with the markers that are used. This possibility could be considered.

j. The phenotype of TFAM is very interesting. If hemocytes are normally involved in the clearance of the apoptotic cells, it suggests that the signals to recruit the hemocytes (ROS?) are disrupted. Further speculation on this phenotype would be of interest.

k. Possible off-targets for the RNAi hits should be acknowledged since only one RNAi line was examined per gene.

l. Review for typos. Figure 1C - lethal is spelled wrong, Figure 3C - "Normale" vs. "normal", Figure 6 - "traduction"?

Point by point response to referees:

Reviewer #1 (Comments to the Authors (Required)):

In this manuscript, the authors performed a RNAi screen to screen for genes that control morphogenetic apoptosis during formation of the leg joint of *Drosophila melanogaster*. They screened more than 1,400 genes (about 10% of the total genome) and identified 182 candidate genes. These were further characterized and grouped into distinct classes that affect specific aspects of morphogenetic apoptosis such as regulators of apoptosis, extrusion and elimination of apoptotic bodies, and acting downstream of apoptosis.

This is a very descriptive study, but the analysis of the candidates was done in a very rigorous manner. In the beginning of the manuscript, leg joint development is well explained which made it easy to follow the mutant analysis. The data are well presented and the conclusions are in line with the data. It will be very interesting in future work to uncover the specific molecular function of these candidate genes for morphogenetic apoptosis.

I have a few questions and suggestions.

1. Which criteria were applied to select the 1,418 genes to be screened for leg joint formation?

Re: The valium 10 collection was the most efficient and with fewer off targets when we started the screen. It was a large collection representing 10% of the genome, which was a reasonable size

for screening a large portion of the genome. This collection was based on a collection of genes expressed in neurons and was described as potentially enriched in transcription factors, ions channels and transporters, however, it seems to have no bias for any specific family of genes (As shown now in FigS5):

2. If I understood the screen and the follow-up analysis correctly, only one RNAi stock was used per gene. Did the authors confirm their findings with additional independent RNAi stocks of the candidate genes? The use of multiple RNAi lines per gene or even better conditional CRISPR/Cas9 knockouts (when available) is standard in Drosophila experimentation.

Re: A second RNAi was used to confirm the results for a large number of candidates giving a phenotype of leg fusion. We use for validation another RNAi from the Valium 20 collection when an RNAi was available. However, this validation step did not concern expected candidates based on the literature (e.g. the gene Ser) nor genes for which a related gene was identified as a candidates (e.g. Arpc4 and Arpc5). In total, 45 candidates out of 100 were validated by another RNAi and 25 out of the 41 final candidates presented in Fig6, which correspond to 60%. All this information is now presented in Table 1.

3. There is some inconsistency in the number of candidate genes: in the text there are 182 candidate genes, in Figure 2A, it's 181 and in Figure 2C, it's 226 candidates. Please clarify and correct where necessary.

Re: Thank you to have noticed this mistake. The correct number is 181 candidates. 226 was including the RNAi giving lethality. For simplification, we remove this class on Fig2C. The text has been corrected and the Fig2C modified accordingly.

4. While the manuscript is well written, I recommend a spell check and proof-reading by a native English speaker. There are several typos and grammatical errors in the text and the figures. For example, in Figure 3 the authors used the word "normale" which does not exist in the English language. In the text, they used the word "raw" instead of "row". There are more examples which I won't list, because the authors didn't provide page and line numbers.

Re: The text has been proof-read by a native English speaker. Thanks for your suggestion.

Reviewer #2 (Comments to the Authors (Required)):

This work by Barbaste and Schott et al., reports the results of a genetic screen to uncover genes involved in Drosophila leg development. The screen was focused on finding genes related to joint development, a process that relies on patterned apoptosis. The screen is well designed, has a very useful readout and more importantly uncovers interesting genes involved not only in induction of apoptosis, but also in mediating the morphogenetic rearrangement downstream of the apoptosis process itself, among other related processes. This is a very useful in vivo model to study apoptosis and how it influences development. The hits identified will be of interest to a wide community as they add on the in vivo regulators of morphogenetic apoptosis. There are some points that would nevertheless strengthen the work and its conclusions:

1. The authors do not mention how penetrant are the phenotypes that they report. It would be important to mention how frequent was a phenotype for it to be considered as representative for a given knockdown condition, even if this was only estimated empirically.

Re: Candidates were selected when at least 50% of the individuals were showing a phenotype. At least 20 individuals were observed for each genetic context and each temperature. When there was not enough adult, we dissected pharates.

This is now mentioned in the M&M.

2. The authors use a distal-less driver to knock down expression of target genes. According to

Flybase, the EM212 driver is also an allele of *dll* and therefore, it not only drives gene silencing of the given RNAi lines, but it also provides a sensitized background. While this does not reduce the value of the screen or of the hits they uncover (in fact is a nice strategy to increase the number of hits) it could be interesting to confirm some of the results with an independent driver (if available). Related to this point, the authors report a great number of genes related to the Notch signaling pathway as hits for their screen and argue that this is due to the central relevance of N pathway in regulating joint development. However, *Dll* seems to regulate *Ser* expression in the legs (Grace Panganiban, John L. R. Rubenstein, 2002) so there could indeed be a genetic interaction with the driver enriching for Notch pathway components. This scenario could at least be mentioned in the discussion section.

Re: Indeed, the *DllGal4-EM212* is a *Dll* mutant. Although it does not show any phenotype in heterozygosis, we add the following sentence in the text to mention that it is possibly a sensitized background that may contribute to the enrichment of Notch related genes in the candidates:

"Of note, the enrichment in Notch related genes could also be due, at least in part, to the fact that the driver used is a Dll hypomorphic mutant that could show some genetic interaction with the Notch pathway."

In addition, we also used another driver at the beginning of the screen, the *Rn-Gal4*, which confirmed our results. This is now mentioned in the text as follow:

"Initially, we used in parallel the RnGal4 driver to test for any driver specific effect. We did this for the first rounds of RNAis tested (119 RNAi). Then, since the phenotypes were similar, we decided to use exclusively the DllGal4-EM212."

3. Figure S3 shows some leg phenotypes in bright field images. Can the authors include a control leg in bright field to have a direct point of comparison?

Re: This has been added.

4. Same figure, some legs do not have hairs (Nipped-B) or have excessive number of hairs (TFAM). This again points for Notch-related phenotypes. It would be useful if these categories could be added and discussed.

Re: Although we agree that this phenotype is interesting, unfortunately, we did not registered hair phenotypes during the screen.

5. Many graphs do not have a title for the y-axis.

Re: This has been modified.

6. In section 2, when the authors describe how they classify the phenotypes, it would be good to give more detailed explanations of each category. Specifically, what is "(2) strong morphological defect", is it any other deformation not included in numbers 3-5?

Re: Indeed, strong morphological defect means important malformation not listed in the other categories. To clarified this point, we modified the text as follow:

“morphological defects in the adult leg (at 25°C, 30°C or both), were classified in the following categories: (1) distal loss (corresponding to a total deletion of the distal part of the leg); (2) misshapen tarsus (corresponding to legs that are correctly differentiated along their proximo-distal axis, but show strong deformations, although no tarsal fusion is observed); (3) tarsal fusions (which correspond to an absence of joint); (4) tarsal fusions combined with other defects (for example tarsal fusion combined with necrotic zones) or (5) strong morphological defect (important malformation not listed in the other categories preventing any visualization of tarsal fusion)”.

7. The authors show that in some conditions, apoptotic bodies are not properly engulfed and cleared. It would be interesting to check if the driver is expressed in macrophages/haemocytes or only in the epithelial tissue of the leg, which could hint to the cell type responsible of the phenotype (neighbouring epithelial cells vs hamocytes).

Re: We agree that it would be interesting to identify the cell type involved in apoptotic bodies clearance defects. Apoptotic fragments are never seen outside the epithelium in control legs and are detected frequently in the neighboring epithelial cells (see Fig5C), strongly suggesting that they are phagocytosed mainly or exclusively by neighboring cells and not by hemocytes. In addition, Dll has not been reported as expressed in hemocytes (flybase). Altogether, this make us propose that the defects of apoptotic bodies accumulation are most probably coming from the epithelial cells, either the dying cell, or the neighboring cells.

This is now commented in the text: *“Of note, these defects in the phagocytosis of apoptotic bodies could be either cell-autonomous (defects in the apoptotic cells themselves) or non-autonomous (defect in the phagocytic cells, which are most probably the neighboring epithelial cells as shown by the frequent detection of apoptotic bodies in epithelial cells from the fold domain, Fig5C)”*

8. Regarding Arp2/3 in the orientation of the folds, it would be worth discussing how boundaries are maintained in place and wether the change of orientation in these knockdowns would be a consequence of misdefinition of the boundary or of a displacement of a correctly patterned boundary.

Re: This is an interesting point. We believe however, that this is out of the scope of this paper. The defect observed in Arp2/3 mutants have been described in one of our recent publications (Martin et al, Dev Cell, 2021; doi: 10.1016/j.devcel.2021.01.005). The misoriented fold phenotype is not affecting the segmental boundaries established by Notch pathway, and even the apoptotic pattern appears unchanged. However, apoptotic forces, instead of being transmitted preferentially in the direction of the future fold, are isotropically distributed in these mutants leading to an increased variability in fold orientation.

9. Same paragraph as above: The reference "Martin et al., 2021" is not in the reference list.

Re: Thank you for pointing out this mistake.

10. The fly drawings used in figure 2A seem to be cropped from another figure and have extra arrows or other characters. The authors may want to use the original vector images found in <https://sites.manchester.ac.uk/fly-facility/training/>

Re: This was due to a bad conversion of the pdf. We apologize for this mistake.

11. Please check the manuscript for typos or minor mistakes, some examples are:
- Short summary: "We carried out" instead of "We realized"
 - Section 1, paragraph 2: ..."activation of Notch (N) signaling in a raw of cells"... row instead of raw.
 - Section 1, paragraph 3: "blue" instead of "bleu"
 - Figure 3B and elsewhere: "normal" instead of "normale"

Re: The text has been proof-read by a native English speaker.

I agree with reviewers 1 and 3 on the use of independent RNAi lines to validate some of the targets or discussing the possibility of off-targets. I also agree it would be important to mention how they define the list of 1,418 genes to be screened. The authors do mention it was an unbiased approach but do not explain how it was defined.

Re: As mentioned above, another Gal4, Rn-Gal4, has been used for a number of RNAi giving similar results. This is now mentioned in the text as follow:

"Initially, we used in parallel the RnGal4 driver to test for any driver specific effect. We did this for the first rounds of RNAi (119 RNAi). Then, since the phenotypes were similar, we decided to use exclusively the DIIGal4-EM212."

Regarding off-target, they are all listed in Bloomington. The vast majority of the lines have no off-targets. The presence of off targets is indicated for each RNAi in Table 3.

We also explain now why the Valium 10 collection has been chosen:

"We chose to use the valium 10 collection of RNAis with known molecular function (excluding CG) since it was a large collection representing 10% of the genome. It was also the most efficient one and with fewer off targets at the time. Finally, while this collection focused on genes expressed in neurons and has been described as potentially enriched in transcription factors, ions channels and transporters (Ni et al, 2009), it seems to have no bias for any specific family of genes."

Reviewer #3 (Comments to the Authors (Required)):

This manuscript describes a screen for genes that affect Drosophila leg morphogenesis, with a focus on events regulated by Notch and apoptosis. They first show that expression of p35 in combination with heterozygosity for H99 leads to morphological defects with a lack of joint formation. They identify nearly 200 genes affecting this process, and categorize them by their effects on Notch and apoptosis. Interestingly, many of them have not been associated with apoptosis before, and the authors propose mechanisms by which cellular mechanics may affect activation of apoptosis or apoptotic cell elimination. All together this is a valuable resource paper that provides a useful starting point for future analysis. No major changes are needed but clarification on a number of points would improve the paper.

The conclusions are strongly supported by the data and no experiments are required. Improvements are suggested below.

Note: the tables cited in the text were not included with the documents for review so they could not be evaluated.

Re: We apologize for this oversight. Tables have been updated (off targets, validation with a 2nd RNAi, with a 2nd driver) and included.

Minor comments:

a. Are the apoptotic cells in the leg disc typically phagocytosed by macrophages or neighboring epithelial cells? More detail on this should be provided (or if it is not known that should be clearly mentioned) since the interpretation of the phenotypes will vary depending on this.

Re: As mentioned above, apoptotic fragments are never seen outside the epithelium in control legs and are detected frequently in the neighboring epithelial cells (see Fig5C), strongly suggesting that they are phagocytosed mainly or exclusively by neighboring cells and not by hemocytes. In addition, Dll has not been reported as expressed in hemocytes (flybase). Altogether, this makes us propose that the defects of apoptotic bodies accumulation are most probably coming from the epithelial cells, either the dying cell, or the neighboring cells.

This is now commented in the text: *“Of note, these defects in the phagocytosis of apoptotic bodies could be either cell-autonomous (defects in the apoptotic cells themselves) or non-autonomous (defect in the phagocytic cells, which are most probably the neighboring epithelial cells as shown by the frequent detection of apoptotic bodies in epithelial cells from the fold domain, Fig5C)”*

b. On p. 4 in the text, it is mentioned that DllGal4 MD23 has a phenotype on its own at 30{degree sign} and refers to Fig S1A'. However, this figure has p35 so it's not clear what the text is referring to.

Re: We agree that the description was misleading. The phenotype of DllGal4 MD23 at 30°C is not shown since it gives rise to a total deletion of the distal part of the leg disc. The one shown is the combination of DllGal4-MD23 with UAS-p35.

“To inhibit apoptosis in the distal part of the leg via Gal4-driven expression of UAS-p35 lines, we first tested the DllGal4-MD23 line, expressed in the Dll expression domain that spread from mid-tibia to the claws and used in previous studies (Manjón et al., 2007; Monier et al., 2015). However, we observed a strong perturbation of Notch signaling in this context, suggesting that DllGal4-MD23 perturb leg patterning (FigS1A’”). In addition, DllGal4-MD23 on its own shows a strong phenotype at 30°C, corresponding to a total deletion of the distal part of the leg disc (not shown).”

c. Are there extra arisal branches in the antenna when apoptosis is inhibited? This phenotype has been previously described for hid loss of function or Diap1 overexpression.

Re: Indeed, a phenotype of extra-branches has been described before in the antenna when apoptosis is inhibited. This phenotype was also frequently observed and registered in our screen. This information has been added to Table 1, which recapitulate all the data.

d. Please include precise genotypes. Is H99 heterozygous in all experiments? Please indicate if the Gal4 or p35 is heterozygous or homozygous.

Re: This is now specified in figure legends.

"In all figures, all the constructs (Gal4, UAS) or mutations (H99) are in heterozygosis."

e. How were the candidate genes selected? It is important to understand if the frequency of "hits" is representative of the genome or if genes important for signaling or patterning were pre-selected.

Re: We chose to screen the valium 10 collection, which was the most efficient RNAi collection and with the fewer off targets when we started the screen. It was a large collection representing 10% of the genome, which was a reasonable size for screening a large portion of the genome without introducing any bias. This is now commented in the text.

f. It is not surprising that Dcp-1 did not give a phenotype since Drice is the major effector caspase. Since mutants of Dcp-1, reaper and hid are viable, these could be examined for leg phenotypes to determine if they should have been identified in the screen. The meaning of the underlined portion of this sentence is not clear: "We noticed however that some known apoptotic key players such as Dcp1 (caspase), skl, rpr and hid (pro-apoptotic genes) do not give any tarsal fusion defects, showing here again the robustness of apoptosis pattern during joints morphogenesis."

Re: We agree that this sentence was misleading since the absence of phenotype could also be due to functional redundancy. We modified this sentence as follow:

"We noticed that some known key apoptotic players such as Dcp1 (caspase), skl, rpr and hid (pro-apoptotic genes) do not give any tarsal fusion defects, which is coherent with the fact that mutants of these genes are viable."

g. They use Dcp-1+ as a read out for apoptosis but Dcp-1 positivity/caspase activity is not always associated with apoptosis. It would be best to confirm it in some other way - for example with DAPI to show that pyknotic nuclei colocalize with the Dcp-1 label, or with TUNEL.

Re: Indeed, Dcp1 is not always related to apoptosis. However, early apoptotic cells were very rare and the vast majority of cells positives for Dcp1 were already fragmented, confirming that they are apoptotic cells. DNA condensation and fragmentation was also clear in the DAPI staining.

h. It is not clear why they assume it is "accumulated apoptosis" (apoptotic cell elimination) rather than increased apoptosis. If they can rule out increased apoptosis they should explain how this was determined.

Re: We agree that "accumulated apoptosis" was an overstatement since we can not be sure that it corresponds to an increase in apoptotic cell number or a defect of apoptotic bodies elimination and accumulation, or both. We rename this class of phenotypes as "Increased apoptosis". We clarify this point as follow:

"The last class of candidates showed an over-abundance or accumulation of apoptotic bodies in the leg discs. Apoptotic corpses are present in abnormally high numbers possibly because more cells are dying by apoptosis, because the corpses are not correctly eliminated, or both. We note however that apoptosis appears normally patterned, taking place in the presumptive fold, rather suggesting a defect in apoptotic bodies elimination, although we cannot totally exclude that some of them correspond to an increase in the number of dying cells."

i. The GFP sensor or Dcp-1 antibody may not be detectable in late phagocytosed corpses that have

acidified. The mutants that don't show a defect in apoptosis could affect these late stages of phagocytosis that are not detectable with the markers that are used. This possibility could be considered.

Re: We agree that a limitation of our screen is that we cannot analyze mutant phenotypes for late phagocytosis and cannot really address the impact of these late steps in the morphogenetic process. It would be interesting to analyze late stage of phagocytosis in our candidates in the future.

j. The phenotype of TFAM is very interesting. If hemocytes are normally involved in the clearance of the apoptotic cells, it suggests that the signals to recruit the hemocytes (ROS?) are disrupted. Further speculation on this phenotype would be of interest.

Re: Since apoptotic bodies are most probably eliminated by neighboring cells, we chose not to comment this point in the manuscript.

k. Possible off-targets for the RNAi hits should be acknowledged since only one RNAi line was examined per gene.

Re: Off-targets are listed in Bloomington. The vast majority of the Valium 10 RNAi lines have no off-targets. The presence of off targets is now indicated for each RNAi in Table 3.

l. Review for typos. Figure 1C - lethal is spelled wrong, Figure 3C - "Normale" vs. "normal", Figure 6 - "traduction"?

Re: The text has been proof-read by a native English speaker.

June 26, 2023

RE: Life Science Alliance Manuscript #LSA-2023-01967-TR

Dr. Magali Suzanne
Centre de Biologie Integrative
118 route de Narbonne
Toulouse 31062
France

Dear Dr. Suzanne,

Thank you for submitting your revised manuscript entitled "Dissecting morphogenetic apoptosis through a genetic screen in *Drosophila*". We would be happy to publish your paper in Life Science Alliance pending final revisions necessary to meet our formatting guidelines.

- Table 2 needs reformatting as pointed out by both Reviewer 1 and 3
- please upload your Tables in editable .doc or excel format;
- please add ORCID ID for the corresponding (and secondary corresponding) author--you should have received instructions on how to do so
- please add the Twitter handle of your host institute/organization as well as your own or/and one of the authors in our system
- please move your main, supplementary figure, and table legends after the references section
- please add a conflict of interest statement to your main manuscript text
- in the manuscript text, there is a callout for Figure 6. However, this is a Graphical Abstract. Please make the necessary correction.
- we encourage you to revise the figure legends for figure S4 such that the figure panels are introduced in an alphabetical order
- please add a callout for Figure 1E to your main manuscript text

Figure checks:

- please indicate the scale bar size in Legends for Figs. 1E, 2B, 3A,B and D, 4A-D, 5D, S1A, S3

A. FINAL FILES:

-- Summary blurb (enter in submission system): A short text summarizing in a single sentence the study (max. 200 characters including spaces). This text is used in conjunction with the titles of papers, hence should be informative and complementary to the title. It should describe the context and significance of the findings for a general readership; it should be written in the

present tense and refer to the work in the third person. Author names should not be mentioned.

B. MANUSCRIPT ORGANIZATION AND FORMATTING:

Sincerely,

Reviewer #1 (Comments to the Authors (Required)):

The authors have addressed all my comments and concerns. I have no further suggestions for improvement, except Table 2 needs reformatting.

Reviewer #2 (Comments to the Authors (Required)):

This work by Barbaste and collaborators shows the results of an RNAi screen designed to identify genes involved in morphogenetic apoptosis that shapes joint formation during leg disc development. The screen identified many genes involved in this process, including cytoskeletal components, Notch-related genes, and as expected, components of the apoptosis pathway. The screen is well designed and validated by independent Gal4 and RNAi lines. Besides pinpointing individual genes involved in morphogenetic apoptosis, it shows the interplay between signaling, mechanics, patterning in apoptosis regulation during morphogenesis. The authors have addressed all the concerns raised in the previous round of revisions.

Reviewer #3 (Comments to the Authors (Required)):

The authors have addressed my concerns, and this provides a very nice resource for the community. Table 2 does not seem to be formatted correctly but everything else is great.

July 13, 2023

RE: Life Science Alliance Manuscript #LSA-2023-01967-TRR

Dr. Magali Suzanne
Centre de Biologie Integrative
118 route de Narbonne
Toulouse 31062
France

Dear Dr. Suzanne,

Thank you for submitting your Resource entitled "Dissecting morphogenetic apoptosis through a genetic screen in *Drosophila*". It is a pleasure to let you know that your manuscript is now accepted for publication in Life Science Alliance. Congratulations on this interesting work.

DISTRIBUTION OF MATERIALS:

Again, congratulations on a very nice paper. I hope you found the review process to be constructive and are pleased with how the manuscript was handled editorially. We look forward to future exciting submissions from your lab.

Sincerely,
